# Novel Thieno [2,3-*b*]pyridine Anticancer Compound Lowers Cancer Stem Cell Fraction Inducing Shift of Lipid to Glucose Metabolism

**DOI:** 10.3390/ijms231911457

**Published:** 2022-09-28

**Authors:** Matij Pervan, Sandra Marijan, Anita Markotić, Lisa I. Pilkington, Natalie A. Haverkate, David Barker, Jóhannes Reynisson, Luka Meić, Mila Radan, Vedrana Čikeš Čulić

**Affiliations:** 1Department of Medical Chemistry and Biochemistry, University of Split School of Medicine, 21000 Split, Croatia; 2School of Chemical Sciences, The University of Auckland, Auckland 1010, New Zealand; 3MacDiarmid Institute for Advanced Materials and Nanotechnology, Wellington 6140, New Zealand; 4School of Pharmacy and Bioengineering, Keele University, Staffordshire ST5 5BG, UK; 5Department of Biochemistry, Faculty of Chemistry and Technology, University of Split, 21000 Split, Croatia

**Keywords:** breast cancer cells, cancer stem cells, newly synthesized thieno [2,3-b]pyridine compound, glycosphingolipids, metabolomics

## Abstract

Due to the role of cancer stem cells (CSCs) in tumor resistance and glycosphingolipid (GSL) involvement in tumor pathogenesis, we investigated the effect of a newly synthesized compound (3-amino-*N*-(3-chloro-2-methylphenyl)-5-oxo-5,6,7,8-tetrahydrothieno[2,3-*b*]quinoline-2-carboxamide **1** on the percentage of CSCs and the expression of six GSLs on CSCs and non-CSCs on breast cancer cell lines (MDA-MB-231 and MCF-7). We also investigated the effect of **1** on the metabolic profile of these cell lines. The MTT assay was used for cytotoxicity determination. Apoptosis and expression of GSLs were assessed by flow cytometry. A GC–MS-coupled system was used for the separation and identification of metabolites. Compound **1** was cytotoxic for both cell lines, and the majority of cells died by treatment-induced apoptosis. The percentage of CSCs was significantly lower in the MDA-MB-231 cell line. Treatment with **1** caused a decrease of CSC IV^6^Neu_5_Ac-nLc_4_Cer+ MDA-MB-231 cells. In the MCF-7 cell line, the percentage of GalNAc-GM1b+ CSCs was increased, while the expression of Gg_3_Cer was decreased in both CSC and non-CSC. Twenty-one metabolites were identified by metabolic profiling. The major impact of the treatment was in glycolysis/gluconeogenesis, pyruvate and inositol metabolism. Compound **1** exhibited higher potency in MBA-MB-231 cells, and it deserves further examination.

## 1. Introduction

Cancer is a leading cause of death in almost every country of the world. Among all cancers, breast cancer is present in 11.7% of new cases in both sexes, while in women the incidence is 24.5%, with a mortality rate of 15.5% [1].

Triple negative breast cancer, TNBC, is an aggressive and heterogeneous disease, defined by the lack of receptors for estrogen (ER), progesterone (PR) and receptor 2 for epidermal factor (HER2), and accounts for 10–25% of all breast cancers. Due to the lack of targeted molecular therapy, this type of cancer is the most aggressive, with the worst prognosis [2,3].

Cancer stem cells (CSC) are a small, proliferating subpopulation of cancer cells that have the ability to self-renew and reform a primary tumor [4].

Breast cancer stem cells (BCSCs), defined by the CD44^+^/CD24^−^ phenotype, show unique characteristics, including tumorigenicity, self-renewal, differentiation, metastatic potential and, most importantly, therapeutic resistance to most antitumor therapies, and it is believed that BCSCs are responsible for tumor recurrence and metastasis [5,6,7,8].

Glycosphingolipids (GSL) are glycolipids that make up the plasma membranes of mammalian cells, but they are also found in intracellular organelles [9]. They consist of hydrophobic ceramides and hydrophilic carbohydrate residues. Based on their basic carbohydrate structures, GSLs are classified into numerous series, including ganglio- and globo-series, which have been analyzed in this study [10].

Because of the diversity of GSLs that form cell membranes, and due to their communication that takes place through these lipid barriers, their biological and chemical properties have been of interest in cancer research for some time. Gangliosides play a key role in embryogenesis, the immune system, cell growth, development and death, tumor progression and pathogen–host interaction. As they are involved in tumor pathogenesis, some gangliosides can be found in tumor tissue, but not in normal healthy tissue [11]. Research shows that ganglioside metabolism in a tumor plays a significant role in its formation and progression [12].

Cellular tumor malignancy is defined by several key phenotypes: apoptosis, cell motility, tyrosine kinase activity for epidermal growth factor receptor, angiogenesis and cell adhesion. Each of these phenotypes is regulated by a different glycosylation status [13].

Increased α2-6 sialylation is associated with increased metastasis and resistance to chemo- and radiotherapy of colorectal cancer [14], and inhibition of the first step of glycosphingolipid synthesis inhibits the colon cancer cell cycle [15].

There are only a few studies about the expression profile of GSL on CSC. Liang et al. reported that the level of Gb3Cer in breast CSC is significantly reduced while GD2, GD3, GM2 and GD1a acidic GSL (gangliosides) are increased in CSCs compared to non-stem breast cancer cells [16]. Levels of glycosyltransferase (ST3GAL5, B4GALNT1, ST8SIA1 and ST3GAL2) mRNAs, involved in those ganglioside syntheses, are increased too. Furthermore, a deletion of the glycosyltransferase-encoding genes for the key enzyme in globo-sides synthesis enhances chemoresistance and increases of number of CD24low/CD44high cells in epithelial cancer cells [17]. Cheung et al. showed that the expression of the enzyme galactosyltransferase (β3GalT5) in breast cancer stem cells and cancer cells is essential for cancer cell survival [18].

The newly synthesized (*E*)-3-amino-5-(3-(3-bromophenyl)acryloyl)-*N*-(3-chloro-2-methylphenyl)-6-methylthieno [2,3-*b*]pyridine-2-carboxamide (Compound) [19] belongs to the group of thieno [2,3-b]pyridine, a family of compounds that were discovered by virtual screening [20], targeting phosphoinositide specific phospholipase C (pi-PLC) and which have been found to have antitumor activity on numerous tumor cell lines [21]. Whilst pi-PLC is a known target, other cellular targets are believed to be involved [22]. Previous examples of these compounds we have produced [23,24,25] have generally utilized a cycloalkyl ring fused at the 5,6-positions of the pyridine ring of the thieno [2,3-b]pyridine to induce improved activity. This increase in activity is proposed to be the result of increased overall lipophilicity and improved interactions of the compounds in a lipophilic pocket of the pi-PLC enzyme. Compound **1** utilized an alternative approach using a tethered propyl-aryl group, which has greater rotation freedom than the cycloalkyl derivatives, to interact with the lipophilic pocket, and resulted in improved antiproliferative activity [19]. Importantly, this class of compound has specifically been shown to inhibit of the growth of triple-negative breast cancer cell lines in nanomolar concentrations [19,26].

Considering the role of CSCs in tumor recurrence, metastasis and drug resistance, we aimed to investigate the effect of a newly synthesized thieno [2,3-*b*]pyridine compound, from the same family of compounds as previously reported [27], on CSC glycosphingolipid expression. The four gangliosides GM2, GalNacGM1b, IV^3^Neu5Ac-nLc_4_Cer and IV^6^Neu5Ac-nLc_4_Cer, and two neutral GSLs (Gg_3_Cer and Gb_4_Cer), were examined. The final step in Gg_3_Cer synthesis is the catalyzation with glycosyltransferase B4GALNT1, of which deletion was shown to cause a phenotype change from CSC to a non-CSC [16]. GSL expression was compared between CSCs and non-CSCs.

Metabolic profiling of cancer cells has a growing role in the elucidation of the molecular basis of cancer development, progression and prognosis. It represents a powerful approach for the characterization of phenotypes and the distinction of specific metabolic states due to environmental alterations. The early observations by Dr Otto Warburg revealed that fundamental metabolic differences exist between malignant tumor cells and adjacent normal cells [28]. The molecular phenotyping of different cancer cell lines and monitoring of their response to different drug treatments can help in the search of new candidate biomarkers and to elucidate new therapeutic targets. These observations suggest that targeting the peculiar metabolic pathways in cancer might be an effective strategy for cancer therapy [29].

In this study, we investigated the effect of a newly synthesized thieno [2,3-*b*]pyridine compound on the metabolic profile of two breast cancer cell lines with different hormone receptor expressions: MDA-MB-231 (triple negative) and MCF-7 (ER+, PR+, HER2-). **A** GC–MS coupled system was used for separation and identification of metabolites.

## 2. Results

### 2.1. Cell Viability

Cells were treated with different concentrations of compound **1** for 24, 48 and 72 h in the MDA-MB-231 (1A) and in the MCF-7 cell line (1B), and cell metabolism was evaluated by the 3-(4,5-dimethylthiazolyl-2)-2,5-diphenyltetrazolium bromide (MTT) assay. Cell viabilities are shown in Figure 1. In the MDA-MB-231 cell line, Compound **1** was shown to be cytotoxic in 0.05 µM concentration after only 24 h of treatment. However, cytotoxicity increased proportionally with increasing concentration and time of treatment. Less than 50% of cells were live after 48 and 72 h of treatment with the concentration 2.5 µM of Compound **1**. Maximal cytotoxicity was achieved for 72 h after treatment with ten times higher concentration (25 µM) of **1**. In the MCF-7 cell line, Compound **1** was slightly less effective, but cytotoxicity increased proportionally with concentration and time of treatment, with less than 50% live cells after 72 h of treatment with concentration 0.1 µM. The maximum cytotoxic effect in this cell line was after 72 h after treatment with 7.5 µM of **1.**

The inhibitory concentration for 50% of cells (IC_50_) for the MDA-MB-231 cell line after 48 h of treatment was 2.082 µM, whilst for the MCF7 cell line it was 2.053 µM.

### 2.2. Apoptosis Rate

To determine the type of cell death induced by 48 h treatment with 2 µM (the average IC_50_ for both cell lines) of compound **1**, we determined early (Annexin-V^+^PI^−^ subpopulation), late (Annexin-V^+^PI^+^ subpopulation) and percentage of total apoptosis. The treatment induced an increase of the percentage of apoptotic cells in both cell lines, as shown in Figure 2. In the MDA-MB-231 cell line, compound **1**-treated cells showed a significant increase in early (A), late (B) and total (C) apoptosis, while in the MCF-7 cell line, the increase was found in late (F) and total (G) apoptosis compared with non-treated cells.

### 2.3. Mammosphere Formation

To determine whether the cancer stem cells are sensitive to compound **1,** the number of mammospheres formed after 7 days of treatment was counted (Figure 3A,B). After treatment with compound **1**, the number of mammospheres was reduced in both cell lines, but significantly only in MCF-7 (Figure 3B). However, the effect of compound **1** was observed in the size and shape of the mammospheres—in both cell lines mammospheres formed in wells with the compound **1** were significantly smaller with an irregular shape (Figure 3C,D).

### 2.4. Cancer Stem Cells

To investigate whether compound **1** contributes to the reduction of the cancer stem cells (CSC) number, we determined CD44^+^/CD24^−^ subpopulation in both the cell lines. Treatment with **1** resulted in a statistically significant decrease of the cancer stem cells in the MDA-MB-231 cell line, known to have a higher CSC population than other breast cancer cell lines (Figure 4A). The number of CSC in the MCF7 cell line also decreased after treatment with **1**, but not in a statistically significant manner (Figure 4B).

### 2.5. Expression of Terminally Sialylated Gangliosides on CSCs and Non-CSCs

Sialylation is involved in cell fate processes during development, reprogramming and tumor progression [30,31]. Therefore, the expression of terminally sialylated gangliosides at CSCs (CD44^+^/CD24^−^ subpopulation) and non-CSCs (CD44^+^/CD24^+^, CD44^−^/CD24^−^ and CD44^−^/CD24^+^ subpopulation; abbreviated as non-CD44^+^/CD24^−^) was studied, with the aim of determining whether the cytotoxic effects of **1** are mediated via different GSL membrane contents. The expression of each GSL per one cell is represented with the geometric mean fluorescence intensity (GMI). The portion of the cells that are GSL positive is an interesting parameter, but it is of less impact in comparison to GMI. Treatment with **1** resulted only in a significant percentage decrease of CSC IV^6^Neu5Ac-nLc_4_Cer-positive cells in the MDA-MB-231 cell line (Figure 5A). Expression of IV^3^Neu5Ac-nLc_4_Cer was not affected by treatment with compound **1** in both cell lines, as shown in Figure 5.

### 2.6. Expression of Gangliosides and Neutral GSLs with Terminal GalNAc Residue on CSCs and Non-CSCs

The expression of these glycosphingolipids was not affected by treatment with compound **1** in the MDA-MB-231 cell line (Figure 6, upper row). In the MCF-7 cell line, the percentage of GalNAc-GM1b-positive cells was significantly increased within the CSC cell subpopulation (CD44^+^/CD24^−^), whilst the percentage of Gg_3_Cer-positive cells was decreased within both subpopulations, CSC and non-CSC (Figure 6, lower row).

### 2.7. Comparative Metabolic Profiling of Breast CSC Using GC–MS

The aim of this study was to compare the metabolic response of MDA-MB-231 and MCF-7 breast cancer cells induced by compound **1**. Metabolic profiling was performed using GC–MS, and 21 metabolites were identified (Table 1). Only compounds listed in the Human Metabolome Database (HMDB 4.0) were selected. Ultimately, entries with more than 50% of missing values were excluded from the study. The objective was to identify the effect of compound **1** on each metabolite and to find metabolites that were significantly different between treated and control cells. The test of statistical significance revealed that the results for MDA-MB-231 cells are more significant than for MCF-7 cells. As listed in Table 1, 6 metabolites were significantly different (*p* < 0.05) from the control in MDA-MB-231 cells and just one metabolite in MCF-7 cells. One-way analysis of variance (ANOVA) was used for statistical significance among MDA-MB-231 and MCF-7 cells. Three significant features (pyruvate, myo-inositol and lactate) were found using the given threshold (*p* < 0.05) for the one-way ANOVA and post hoc Tukey test. PCA results showed that compound **1** treatment causes metabolic alternations in both cell lines. PCA score plots (Figure 7) visualize the clustering of treated group versus control. PC1 accounts for 67% and 50% of the variation in the data in MDA-MB-231 and MCF-7 line, respectively. The score plots showed that the control group and treated group could be clearly distinguished.

Quantitative enrichment analysis was employed to identify patterns of metabolite concentrations and to help elucidate possible biological mechanisms. Results were considered statistically significant with *p* < 0.05, and no statistical difference was established for MCF-7 cells. The results of the analysis indicate that treatment had a major impact in glucose/energy metabolism, specifically in glycolysis/gluconeogenesis, the pyruvate metabolism, Warburg effect and inositol metabolism (Figure 8).

## 3. Discussion

It is well known that cancer is linked to other chronic medical conditions [34]. For example, patients diagnosed with metabolic syndrome show an increased incidence and aggressiveness of tumor formation [35]. In this study, metabolic profiling of two different breast cancer cell lines revealed that compound **1** increased pyruvate and decreased lactate, phosphate, myo-inositol, glycerol monopalmitate and glycerol monostearate in MDA-MB-231 cells. An inverse effect was noticed for two metabolites in MCF-7 cells: increased lactate and glyceryl-monostearate (Figure 9).

Compound **1** was more cytotoxic to MDA-MB-231 cells, and the results showed that it shifted glycolysis to the aerobic pathway (increased pyruvate, decreased lactate), as seen by the significant reduction in the number of cancer stem cells, which prefer anaerobic glycolysis. The opposite could be observed in MCF-7 cells, where the compound increased lactate, indicating a shift to anaerobic glycolysis. The decrease of phosphate after compound **1** treatment in MDA-MB-231 cells could be caused by the increased binding of phosphate ions to their transporters in the inner mitochondrial membrane during aerobic glycolysis. The binding of phosphates to its transporters is favored with hydrogen ions present in intermembrane mitochondrial space during aerobic glycolysis.

Inverse compound **1** effect upon the glycerol-monostearate concentration in MDA-MB-231 in relation to MCF-7 cells could be developed due to different growth factor receptors, present in two cell lines. Genes involved in lipid metabolism are highly enriched in diverse models of cellular transformation and are thereby linked to cancer [34]. These genes are for OLR1, oxidized low-density lipoprotein receptor 1; SNAP23, crucial protein for storage of neutral lipids in lipid droplets and translocation of GLUT4 to the plasma membrane [36]; VAMP4, protein-driving lipid droplet forming, together with SNAP23; SCD1, stearoyl-CoA desaturase-1; and SREBP1, master lipogenesis regulator, sterol regulatory element binding protein 1. SREBP1 is targeted to inhibit lipid metabolism, thereby obstructing epithelial–mesenchymal transition in breast cancer [37]. Fatty acid synthase plays an important role in cancer pathogenesis [38]. Growth factor-dependent or independent autophosphorylation of receptor tyrosine kinases leads to signal transduction cascades. MCF-7 cells express estrogen, progesterone and androgen receptors while MDA-MD-231 cells express only androgen receptors [39]. However, progesterone can transmit its signal in MDA-MB-231 cells by binding to progesterone receptor membrane component 1, which has been shown to regulate some cancer hallmarks [40]. Steroid hormones including estradiol, progestins and androgens trigger transduction mechanisms that ultimately stimulate fatty acid synthase expression through the modulation of the transcription factor sterol regulatory element-binding protein 1c (SREBP1c), which binds to and activates sterol regulatory elements in the promoter region of fatty acid synthase [38]. The lowering of glycerol monopalmitate and glycerol monostearate in MDA-MB-231 cells by compound **1** indicates its influence upon the reversal of lipid to glucose metabolism. This shift includes the inhibition of the pentose phosphate pathway, which is the source of reductive power (NADPH) for palmitate and stearate synthesis, and the inhibition of citrate transport from mitochondria in the cytosol [38]. Fatty acid synthesis requires cleavage of the citrate in cytosol to yield acetyl-CoA, the precursor of palmitate and stearate.

Compound **1** treatment did not influence glycosphingolipid expression per one cell, but the percentage of IV^6^NeuAc-nLc_4_-positive cells was decreased in MDA-MB-231 CSC, while MCF-7 CSC involved increased GalNAc-GM1b+ and decreased Gg3Cer+ subpopulation percentage. Gg3Cer+ subpopulation was also decreased within the non-CSC of the MCF-7 cell line, which is in accordance with finding of Liang et al. After the knockdown of ST8SIA1 and B4GALNT1, they found significantly reduced expression of GD2 and GD3 gangliosides and phenotype change from CSC to a non-CSC [16]. Namely, Gg3Cer-neutral GSL is also synthesized with B4GALNT1 (Figure 10). Terminally α2-6-sialylated gangliosides of the neolactose series IV^6^NeuAc-nLc_4_ are recognized earlier as receptors for rViscumin, which is used as an adjuvant cancer therapy [41]. Recently, we described increased IV^6^NeuAc-nLc_4_ expression in both CSCs and non-CSCs of the MDA-MB-231 cell line after treatment with a different compound that belongs also to the thieno [2,3**-***b*]pyridine group of molecules [27]. Gg3Cer has been detected in murine lymphosarcoma cell lines [42] and in myeloid leukemia cells as major neutral glycosphingolipid [43].

The finding of an increased GalNAc-GM1b+ and decreased Gg3Cer+ subpopulation percentage in MCF-7 CSC indicates that the addition of neuraminic acid (Neu5Ac) and two sugar residues (GalNAcβ1-4Galβ1) to Gg3Cer favors the survival of MCF-7 CSC (Figure 10).

The gene-coding GalNAc-transferase (GALNT2-SRD5A3) that binds glycosyl residues to glycoproteins belongs to a family of genetic alteration-driven genes and drug target genes acting as prognostic biomarkers in breast cancer [45]. GALNT2 gene-coding O-GalNAC transferase 2, which synthesizes Tn-antigen in core O-glycosylation (mucin type) glycoproteins, is associated with high cholesterol and triglyceride levels [46]. Investigation by Pucci et al. point to GALNT2 as one of the most consistent malignancy-associated enzymes together with B4GALNT1, B4GALT5, B3GNT5 and ST3GAL2 [44]. B4GALT5 and B4GALNT1 are involved in steps of Gg3Cer (called also asialo GM2) and GalNAc-GM1b synthesis, as shown in Figure 10.

GalNAc-GM1b is a ganglioside, acidic glycosphingolipid that contains sialic acid. ST3GAL2 catalyzes sialic acid transfer to Gg4Cer (called also asialo GM1). GalNAc-GM1b is described as minor in four human uterine cervical carcinoma-derived cell lines [47]. Sphingosine and ceramide (Figure 10) can be composed of different fatty acids—for example, stearic acid—with 18 carbon atoms [48], but mostly from palmitic acid (C16), which is the major fatty acid present in metabolism [49]. Recently, Cruz et al. showed beneficial effects of ceramide use in nanomedicine, targeting it in combination with doxorubicin, within liposomes, to MDA-MB-231 cells [50]. MCF cells are able to escape from the ceramide-related apoptosis by the activation of ganglioside synthesis [51]. In our study, compound **1** significantly increased early and late apoptosis in MDA-MB-231 cells, but only late apoptosis in MCF-7 cells. Only MCF-7 CSC showed an increased percentage of ganglioside GalNac-GM1b+ cells. We can assume that those cells contained lower ceramide levels and are thereby less sensitive to early apoptosis. Again, as well as for the aforementioned Gg3Cer, there is accordance of the GalNac-GM1b finding with the results of Liang et al. [16]. They found an increased level of mRNA coding ST3GAL2 glycosyltransferase, involved in GalNac-GM1b synthesis, in CSCs compared to non-stem breast cancer cells (Figure 10).

In conclusion, compound **1** exhibited higher potency in MBA-MB-231 cells, the triple negative breast cancer cell line, and it deserves further examinations in in vitro and in vivo models.

## 4. Materials and Methods

### 4.1. Chemistry and Cell Line

(*E*)-3-Amino-5-(3-(3-bromophenyl)acryloyl-*N*-(3-chloro-2-methylphenyl)-6-methylthieno [2,3-*b*]pyridine-2-carboxamide (compound **1**) (Figure 11) was prepared as reported [19] and dissolved in a minimum amount of dimethyl sulfoxide (DMSO) prior to dilution in buffer. Cancer cell lines MDA-MB-231 and MCF-7 were grown in a humidified incubator at 37 °C and 5% CO_2_ in Dulbecco’s Modified Eagle Medium (DMEM, Sigma-Aldrich, Steinheim, Germany), enriched with 10% fetal bovine serum and 1% antibiotics. 

### 4.2. Cytotoxic Activity Assay

To estimate cell viability, MTT was assessed [52]. Equal numbers of cells were seeded in three replicates. After overnight incubation, cells were treated with complete media or individual solutions of **1** at 0.05, 0.1, 0.25, 0.5, 1, 2.5, 5, 7.5, 10 and 25 µM in complete media, in three repetitions, for 4, 24, 48 and 72 h. Following treatment, cells were incubated with 0.5 mg/mL MTT in media for 2 h, after the media was ousted and DMSO was added. Absorbance was measured at 570 nm [53].

### 4.3. Flow Cytometric Analyses

Equal numbers of cells were plated in 6-well plates and treated with 2 µM **1** and then analyzed for apoptosis and glycosphingolipid expression. Subsequently, the cells were trypsinized, collected and washed twice with phosphate buffered saline (PBS), resuspended in 100 µL of the binding buffer containing 5 µL Annexin-V-FITC and 5 µL of PI (Annexin-V-FITC Apoptosis Detection Kit I, BD Biosciences, San Jose, CA, USA), incubated for 15 min at room temperature in the dark and analyzed by flow cytometry (BD Accuri C6, BD Biosciences, San Jose, CA, USA). Using the FlowLogic Software (Inivai, Mentone, VIC, Australia), the percentages of apoptotic cells (Annexin-V-positive cells) were determined and presented as mean ± standard deviation (SD).

Both cell lines treated with **1** for 48 h, as well as the controls, were stained with anti-CD44-FITC (BD Biosciences, San Jose, CA, USA), anti-CD24-phycoerythrin (PE, eBioscience, Inc., San Diego, CA, USA) and anti-GSL antibodies. The primary antibodies against Gb_4_Cer, IV^3^Neu5Ac-nLc_4_Cer, IV^6^Neu5Ac-nLc_4_Cer, GM2, Gg_3_Cer (gangliotriaosylceramide, GalNAcβ1-4Galβ1-4Glcβ1-1Cer) and GalNAc-GM1b were chicken polyclonal antibodies being produced and characterized by the laboratory of Dr. J. Müthing [54].

Secondary antibodies conjugated with eFluor 660 fluorochrome (Abcam, Cambridge, UK) were used for detection of the binding of primary anti-GSL antibodies. 

Data acquisition was performed on a BD Accuri C6 cytometer and analyzed with FlowLogic Software. CD44^+^CD24^−^ cells were gated and CSC were determined. Glycosphingolipids Gb_4_Cer, IV^3^Neu5Ac-nLc_4_Cer, IV^6^Neu5Ac-nLc_4_Cer, GM2, Gg_3_Cer and GalNAc-GM1b were determined on CSCs (CD44^+^CD24^−^) and non-CSCs (non CD44^+^CD24^−^) in both cell lines.

### 4.4. Mammosphere Forming Assay

Cells derived from MDA-MB-231 and MCF-7 cell lines were seeded in 6-well low attachment suspension culture plates (Corning^®^ Costar^®^ Ultra-Low Attachment Multiple Well Plate, Thermo Fisher Scientific, Waltham, MA, USA) at a density of 3.5 × 10^4^ viable cells/well. Cells were grown in 2 mL MammoCult Medium Human Kit, enriched with Proliferation Supplement 0.1 mg/mL, Heparin Solution 4 µg/mL, Hydrocortisone Stock Solution 0.48 µg/mL (all StemCell Technologies, Vancouver, BC, Canada) and antibiotics (1% penicillin/streptomycin, Sigma-Aldrich, Steinheim, Germany) for control, and with addition of 2 µM **1** in enriched media for tested cells. Mammospheres larger than 50 μm were counted after 7 days of incubation using a Motic AE31E Inverted Microscope (Thermo Fisher Scientific, Waltham, MA, USA). 

### 4.5. Quenching, Harvesting and Extraction of Cells for Metabolite Analysis

Metabolic extraction solvents were of LC–MS grade. Methanol and chloroform were obtained from Gram-Mol, Zagreb, Croatia. Reagents for derivatization included methoxyamine hydrochloride, pyridine and N-Methyl-N-(trimethylsilyl) trifluoroacetamide, and ribitol was used as an internal standard (all Sigma Aldrich, Steinheim, Germany). 

### 4.6. Sample Collection for Metabolite Analysis

Cells were grown for 48 h in 6-well plates culture flask to near confluence. The culture medium was discarded, and cells were gently washed with 2 mL PBS and then quenched using cold methanol (3 + 1 mL). In an attempt to effectively quench the metabolism of cells, combined quenching and extraction step was performed, without cell scraping. 20 μL of ribitol (Sigma Aldrich, Steinheim, Germany) was added as an internal standard, and the sample was dried in a vacuum rotational evaporator (Laborota 4000-efficient, Heidolph, Schwabach, Germany).

### 4.7. Sample Derivatization and GC–MS Analysis

The derivatization process involved addition of 25 μL solution consisting of 20 mg/mL methoxyamine hydrochloride in pyridine, followed by constant shaking for 60 min at 50 °C and the addition of MSTFA (N-Methyl-N-(trimethylsilyl)trifluoroacetamide, Sigma Aldrich, Steinheim, Germany) + 1% TMCS (Trimethylchlorosilane, Sigma Aldrich, Steinheim, Germany), with incubation at 50 °C for 30 min for complete derivatization. The sample was dissolved in 100 μL pyridine.

The samples were analyzed using an Agilent 8890 GC system coupled with triple quad spectrometer system MS 7000D GC/TQ. The column was HP-5 MS (30m × 0.25 mm × 0.25 μm, Agilent, Santa Clara, CA, USA) with an oven program set on 60 °C maintained for 2 min, then increased to 210 °C at rate of 10 °C/min, ramped to 240 °C at rate of 5 °C/min and then ramped to 315 °C at a rate of 25 °C/min, and then held at 315 °C for 3 min.

### 4.8. GC-MS Data Preprocessing and Statistical Analysis

Agilent MassHunter Qualitative Analysis software was used for spectral processing (including peak picking, alignment, annotation and integration). Metabolites were identified using the NIST library. The intensity value for each metabolite was normalized to ribitol internal standard signal. A total of 21 metabolites were identified in cell culture samples.

MetaboAnalyst, a platform dedicated to metabolomics data analysis, was used to establish a panel of differentially expressed metabolites across these two cell lines. 

Statistical significance analysis was performed using *t*-test. One-way ANOVA was used to compare the effect of **1** on both cell lines. Principal component analysis (PCA) was used as an unsupervised clustering method. Metabolic set enrichment analysis (MSEA) was employed to establish a link between metabolic fingerprints and metabolite concentration changes.

The IC_50_ values were calculated using GraphPad Prism 7.0 (San Diego, CA, USA). For statistical analyses of apoptosis and glycosphingolipids expression, t-test with unequal variances, one-way ANOVA followed by post hoc Tukey test or Kruskal–Wallis, followed by Dunn’s post hoc test, were performed using statistical software GraphPad Prism 7.0 (San Diego, CA, USA), with the significance set at *p* < 0.05.

## Figures and Tables

**Figure 1 ijms-23-11457-f001:**
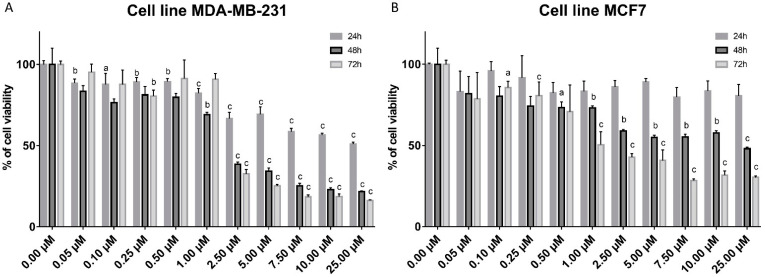
Cell viability after compound **1** treatment. *Notes:* Cells were treated with a different concentration of compound 1 for 24, 48 and 72 h in the MDA-MB-231 (**A**) and in the MCF-7 cell line (**B**), and cell metabolism was evaluated by the 3-(4,5-dimethylthiazolyl-2)-2,5-diphenyltetrazolium bromide (MTT) assay. Data are expressed as a mean from the experiment performed in triplicate ± SD. Columns, mean of viable cells; bars, SD (standard deviation); a, *p* < 0.05; b, *p* < 0.01; c, *p* < 0.001. SD, standard deviation.

**Figure 2 ijms-23-11457-f002:**
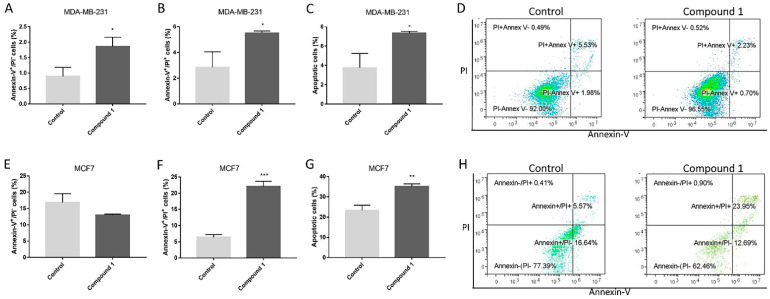
Apoptosis after compound **1** treatment. *Notes:* Percentage and dot plots of early, late and total apoptotic cells without and with **1** treatment for 48 h in the MDA-MB-231 (**A**–**D**) and in the MCF-7 cell line (**E**–**H**). Data represented are expressed as a mean from experiment performed in triplicate ± SD. Columns, mean of cells; bars, SD; * *p* < 0.05; ** *p* < 0.01 *** *p* < 0.001. SD, standard deviation.

**Figure 3 ijms-23-11457-f003:**
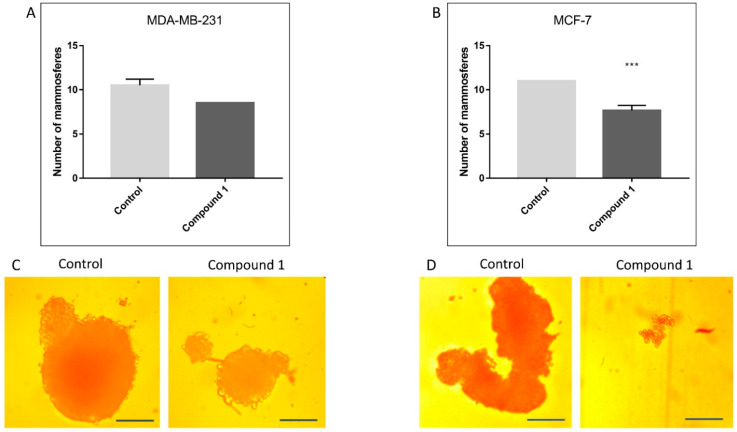
Mammosphere formation after compound **1** treatment. *Notes:* Number of mammospheres without and with compound **1** treatment for 7 days in the MDA-MB-231 (**A**) and in the MCF-7 cell line (**B**), and photos with ×100 magnification (scale bar, 200 μm) in the MDA-MB-231 (**C**) and in the MCF-7 cell line (**D**). Mammospheres with a size over 50 μm were evaluated. Data represented are expressed as a mean from experiment performed in triplicate ± SD. Columns, mean of cells; bars, SD; *** *p* < 0.001. SD, standard deviation.

**Figure 4 ijms-23-11457-f004:**
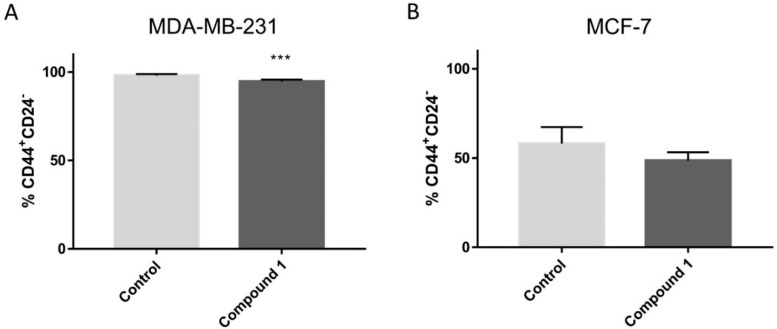
CSCs after compound **1** treatment. *Notes:* Percentage of CD44^+^CD24^−^ CSCs after treatment with compound **1** for 48 h in MDA-MB 231 (**A**) and in the MCF-7 cell line (**B**). Data represented are expressed as a mean from experiment performed in triplicate ± SD. Columns, mean of cells; bars, SD; *** *p* < 0.001. *CSCs* cancer stem cells; SD, standard deviation.

**Figure 5 ijms-23-11457-f005:**
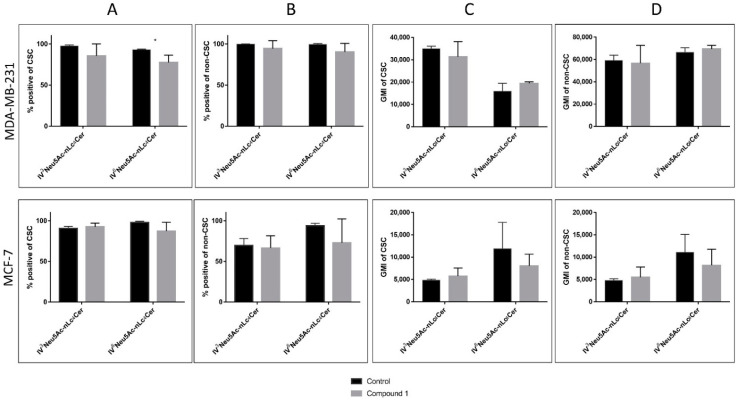
Percentage and geometric mean fluorescence intensity of terminally sialylated ganglioside-positive cell subpopulations. *Notes:* Percentage of CSCs in the MDA-MB-231 and in the MCF-7 (**A**), and non-CSCs in MDA-MB 231 and in the MCF-7 (**B**). Geometric mean fluorescence intensity of CSCs in MDA-MB 231 and in the MCF-7 (**C**), and non-CSCs in the MDA-MB-231 and in MCF-7 cell lines (**D**) after treatment with compound 1 in duration of 48 h. Data are expressed as a mean from experiment performed in triplicate ± SD. Columns, mean of viable cells; bars, SD; * *p* < 0.05. CSC, CD44^+^/CD24^−^ cells; non-CSC, CD44^+^/CD24^+^, CD44^−^/CD24^−^ and CD44^−^/CD24^+^ cells; GMI, geometric mean fluorescence intensity; SD, standard deviation. The designation of the gangliosides follows the IUPAC-IUB recommendations [32] and the nomenclature of Svennerholm [33]: Neu5Ac, *N*-acetylneuraminic acid; nLc4Cer, neolactotetraosylceramide; IV^3^Neu5Ac-nLc4Cer; IV^6^Neu5Ac-nLc4Cer.

**Figure 6 ijms-23-11457-f006:**
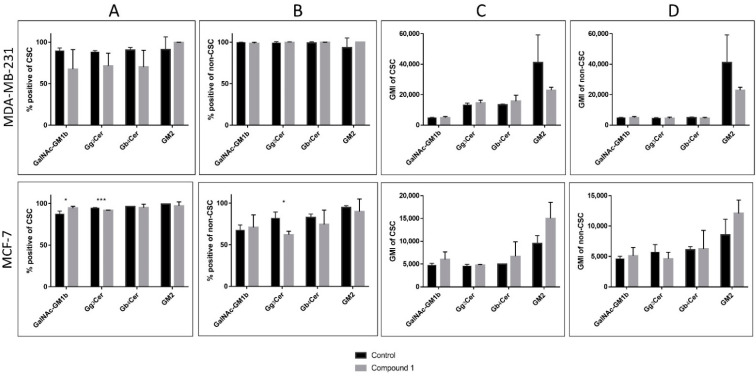
Percentage and geometric mean fluorescence intensity of cell subpopulations positive for ganglioside and neutral GSLs with terminal GalNAc residue. *Notes*: Percentage of CSCs in the MDA-MB-231 and in the MCF-7 (**A**), and non-CSCs in MDA-MB 231 and in the MCF-7 cell lines (**B**). Geometric mean fluorescence intensity of CSCs in MDA-MB 231 and in the MCF-7 (**C**), and non-CSCs in the MDA-MB-231 and in MCF-7 cell lines (**D**) after treatment with compound 1 in duration of 48 h. Data are expressed as a mean from experiment performed in triplicate ± SD. Columns, mean of viable cells; bars, SD, * *p* < 0.05, *** *p* < 0.001. CSC, CD44^+^/CD24^−^ cells; non-CSC, CD44^+^/CD24^+^, CD44^−^/CD24^−^ and CD44^−^/CD24^+^ cells; globotetraosylceramide or Gb4Cer, GalNAcβ1-3Galα1-4Galβ1-4Glcβ1-1Cer; gangliotriaosylceramide or Gg3Cer, GalNAcβ1-4Galβ1-4Glcβ1-1Cer; GM2, II3Neu5AcGg3Cer; GalNAc-GM1b, IV3Neu5Ac-Gg5Cer; GM1b, IV3Neu5Ac-Gg4Cer; gangliotetraosylceramide or Gg4Cer, Galβ1-3GalNAcβ1-4Galβ1-4Glcβ1-1Cer; GMI, geometric mean fluorescence intensity; SD, standard deviation.

**Figure 7 ijms-23-11457-f007:**
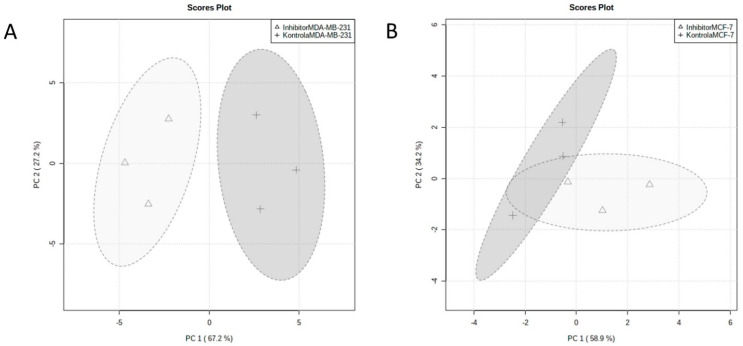
PCA of metabolic profile of MDA-MB-231 and MCF-7 cells. *Notes:* PCA of metabolic profile of MDA-MB-231 (**A**) and MCF-7 cell lines (**B**) after treatment with compound 1 in duration of 48 h. PCA, principal component analysis.

**Figure 8 ijms-23-11457-f008:**
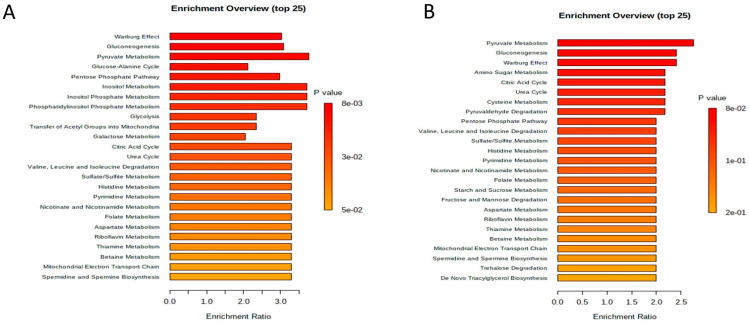
Metabolomic set enrichment analysis of MDA-MB-231 and MCF-7 cells. *Notes:* Metabolomic set enrichment analysis showing the most changed functional metabolism of MDA-MB-231 (**A**) and MCF-7 cell lines (**B**) after treatment with compound 1 in duration of 48 h. Molecular pathways were ranked according to the probability (*p*-value) of finding a particular number of significantly changed metabolites in the compound list of a certain pathway.

**Figure 9 ijms-23-11457-f009:**
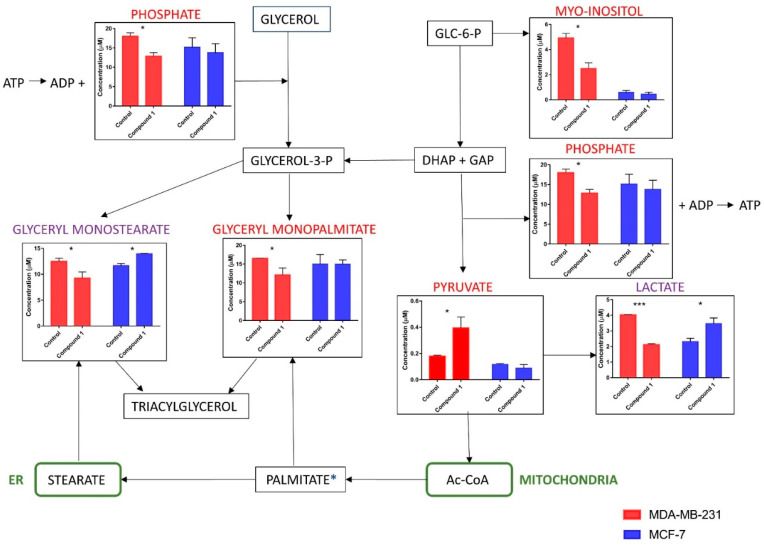
The relative concentrations of the metabolites. *Notes:* The relative concentrations of the metabolites involved in the glycolytic pathway and triacylglycerol pathway (part) in MDA-MB-231 cells (red) and MCF-7 cells (blue) after treatment with compound 1 in duration of 48 h. Data are expressed as the mean value ± SD of three independent experiments. Columns, mean of cells; bars, SD; * *p* < 0.05; *** *p* < 0.001; SD, standard deviation; purple text indicates metabolites concentrations changed after treatment in both cell lines; ER, endoplasmatic reticulum; ADP, adenosine diphosphate; ATP, adenosine triphosphate; GLC-6-P, glucose-6-phosphate; glycerol-3-P, glycerol-3-phosphate; DHAP, dihydroxyacetone phosphate; GAP, glyceraldehyde 3-phosphate; Ac-CoA, acetyl-coenzyme A.

**Figure 10 ijms-23-11457-f010:**
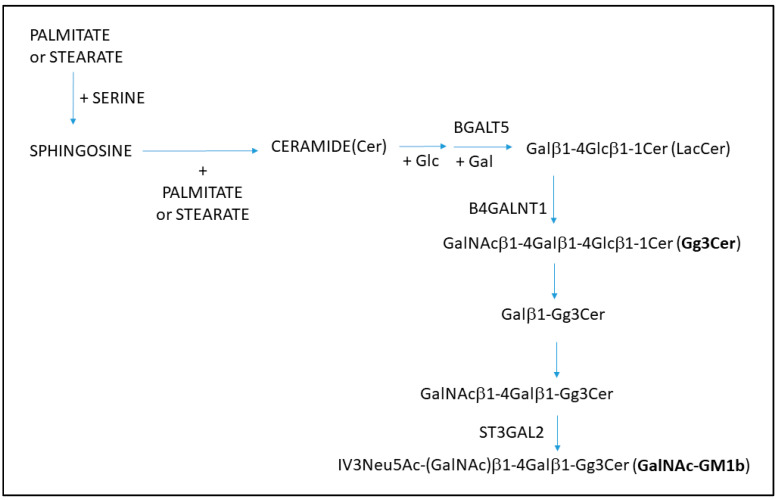
Pathway of Gg3Cer and GalNAc-GM1b glycosphingolipid synthesis in MDA-MB-231 cells. The designation of the glycosphingolipids and their core structures follow the IUPAC-IUB recommendations [32]. Only transferases B4GALNT1, B4GALT5 and ST3GAL2, which are among the most consistently malignancy-associated enzymes [44], are marked at corresponding steps of synthesis. The relative concentrations of the metabolites.

**Figure 11 ijms-23-11457-f011:**
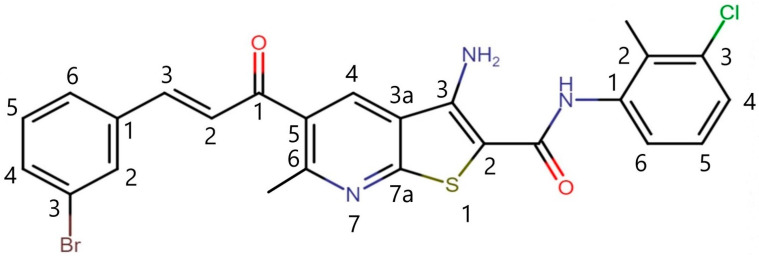
The structure of the newly synthesized anticancer agent (compound **1**). *Note:* Compound **1**, (*E*)-3-amino-5-(3-(3-bromophenyl)acryloyl)-*N*-(3-chloro-2-methylphenyl)-6-methylthieno [2,3-*b*]pyridine-2-carboxamide.

**Table 1 ijms-23-11457-t001:** List of identified compounds in the MDA-MB-231 and MCF-7 cell line.

No.	RT (min)	Metabolite	MDA-MB-231		MCF-7	
			*p* Value	Fold Change	*p* Value	Fold Change
**1**	6.1	Pyruvic acid	0.04 *	2.19	0.15	0.76
**2**	6.3	L-Lactic acid	<0.001 *	0.53	0.03 *	1.50
**3**	7.5	Oxalic acid	0.8	1.09	0.1	0.56
**4**	9.2	Serine				
**5**	9.4	Threonine				
**6**	9.5	Phosphate	0.03 *	0.71	0.5	0.91
**7**	9.9	Glycine	0.28	0.44	0.3	0.63
**8**	11.5	D-Alanine				
**9**	12.4	Pyroglutamic acid				
**10**	12.8	L-Proline	0.1	0.75	0.76	0.89
**11**	16.3	Erythrose	0.1	0.69		
**12**	16.4	4-Hydroxycinnamic acid	0.08	1.22	0.1	1.09
**13**	16.9	D-Fructose	0.3	0.52		
**14**	17.2	D-Glucose	0.5	0.96	0.4	0.88
**15**	17.4	Mannose	0.9	1.07	0.7	0.96
**16**	18.5	Palmitic acid	0.6	0.80	0.4	0.8
**17**	19.5	Myo-inositol	0.03 *	0.71	0.3	0.76
**18**	19.9	Octadecanol	0.4	0.99	0.4	1.07
**19**	21.0	Stearic acid	0.3	0.71	0.5	1.42
**20**	25.8	Glyceryl monopalmitate	0.05 *	0.73	0.9	0.99
**21**	28.5	Glyceril monostearate	0.04 *	0.74	0.02 *	1.2

*Notes:* The intensity value for each metabolite normalized to ribitol internal standard signal in MDA-MB-231 cells and MCF-7 cells after treatment with compound **1** in duration of 48 h. Fold change represents the ratio of the mean value of signal intensity (of three independent experiments) of treated over non-treated cells. *p* values are obtained by Student t test (features with > 50% missing values were excluded); RT (retention time); * statistically significant change.

## Data Availability

The datasets used and/or analyzed during the current study are available from the corresponding author on reasonable request.

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
