# Peer review of "Novel Thieno [2,3-b]pyridine Anticancer Compound Lowers Cancer Stem Cell Fraction Inducing Shift of Lipid to Glucose Metabolism"

_ijms, 2022, doi:10.3390/ijms231911457_

Round 1
Reviewer 1 Report
This article is very concise.... which can be a good thing, but here, I often wish for more details, even in the introduction. For example, cancer stem cells are simply introduced as CD44+ - CD24- cells. That's a bit of a stretch and would benefit from further elaboration, and fine-tuning, especially as complex stem cell markers (gangliosides) are used in this manuscript for cell identification. They also pop up further below, maybe put these paragraphs together.
Then, we may not want to see the chemical synthesis of the compound in focus here, but you may expect a bit more background concerning its properties, relation to other compounds, etc etc., what the novelty here (apart from the fact the compound is "newly synthesized" itself). There is a bit more background on gangliosides, but the functional link to cancer stem cells and their biology is missing. In other words, the intro is jumpy, and disconnected, there is no straight line narrative and that can be improved.
Results: the authors have used a broad spectrum of concentrations of the drug, on both cell lines; it should therefore be possible to calculate a proper EC50 estimation.
Apoptosis: I think the compounds are not very effective at inducing apoptosis, and they don't have to. So maybe this should not be overly discussed.
Mammosphere formation: It's not clear what we gain from the mammosphere assay. It appears to me that the compound is simply blocking proliferation, either specifically or in a non-specific fashion, due to cytotoxicity. The mammosphere assay doesn't add anything to this since they are still growing... i.g., they are proliferating and the drug is clearly antiproliferative (but not so much pro-apoptotic). However, I think it would be much more interesting if the authors would take "mature", fully formed mammospheres, and see if the drug has any long(er) term effects on these aggregates. Simply, since there should be a (marginal, but measurable) enrichment of "stemness" in these spheroids and this may be targeted - as the authors claim - by the drug. They may also have a look at the cells in the centre of these spheroids, as these display the highest degree of "stemness", for whatever reasons.
Cancer Stem Cells: the authors claim in Fig. 4 to look at stem cell populations in these 2 cell lines. I am not convinced these really are stem cells, and would expect additional proof. Are these cells responding to any estrogens? Growth factors (they shouldn't). Are they proliferating (they should be slowly). This requires more detailed analysis and also more explanation. The isolation and characterization of these cells from the 2 cell lines are barely described at all, certainly, no details whatsoever are provided in the results section. What's the ratio of these "stem cells" to luminally differentiated cells? The current paragraph and Fig. 4 cannot be accepted in this format. Any characterization is also missing in the materials & methods part. This is a major problem of the manuscript and MUST be fixed before it can be considered for publication.
Fig. 5: the expression of gangliosides also makes more sense if the ratio of CSCs over "other" cells is shown. At the present moment, I also don't see any interesting and significant effects of the compound. And the small changes that we do see, I don't know what do we learn from those data? So, what's the point?
Maybe the most solid data are shown in Fig. 9 and 10, plus table 1. I have the hunch the authors are more experts in this field than cell biology and cell-based models. But that's okay. Maybe the manuscript should be focused on these metabolic/metabolomic studies, and conclusions should be drawn from these data, not from (maybe) stem cells that aren't properly characterized. I also don't really see the link a) between the compound and stem cell biology, and b) between the compound and the gangliosides. That's really not convincing.
Author Response
Point 1: This article is very concise.... which can be a good thing, but here, I often wish for more details, even in the introduction. For example, cancer stem cells are simply introduced as CD44+ - CD24- cells
Response 1: This phenotype is often used as CSC phenotype. We have added two more recent references on this topic (line 53):
Batlle E, Clevers H. Cancer stem cells revisited. Nat Med. 2017 Oct 6;23(10):1124-1134. doi: 10.1038/nm.4409. PMID: 28985214.
Gaio E, Conte C, Esposito D, Reddi E, Quaglia F, Moret F. CD44 Targeting Mediated by Polymeric Nanoparticles and Combination of Chlorine TPCS2a-PDT and Docetaxel-Chemotherapy for Efficient Killing of Breast Differentiated and Stem Cancer Cells In Vitro. Cancers (Basel). 2020 Jan 23;12(2):278. doi: 10.3390/cancers12020278. PMID: 31979218; PMCID: PMC7072409.
Point 2: That's a bit of a stretch and would benefit from further elaboration, and fine-tuning, especially as complex stem cell markers (gangliosides) are used in this manuscript for cell identification.
Response 2: Gangliosides are not stem cell markers used in our research for cell identification. The aim of our study was to investigate the expression of gangliosides on cancer stem cells and non-stem cells, respectively.
Point 3: They also pop up further below, maybe put these paragraphs together.
Response 3: We put the paragraph discussing glycosphingolipids after the paragraph discussing cancer stem cells (now it starts from line 55).
Point 4: Then, we may not want to see the chemical synthesis of the compound in focus here, but you may expect a bit more background concerning its properties, relation to other compounds, etc etc., what the novelty here (apart from the fact the compound is "newly synthesized" itself).
Response 4: Thank you for your comment. We added few sentences and references discussing background of the compound synthesis in relation to its property and effects relating to other compounds from the same family of compounds (now lines 108 – 118).
Point 5: There is a bit more background on gangliosides, but the functional link to cancer stem cells and their biology is missing.
Response 5: We added three more references discussing the role of glycosphingolipid expression on CSC in tumor malignancy, metastasis and resistance, both to the Introduction and Discussion sections (now lines 74 – 103, 360 - 363 and 401 - 404).
Liang YJ, Ding Y, Levery SB, Lobaton M, Handa K, Hakomori SI. Differential expression profiles of glycosphingolipids in human breast cancer stem cells vs. cancer non-stem cells. Proceedings of the National Academy of Sciences of the United States of America. 2013;110(13):4968-73. Epub 2013/03/13.
Cheung SK, Chuang PK, Huang HW, Hwang-Verslues WW, Cho CH, Yang WB, et al. Stage-specific embryonic antigen-3 (SSEA-3) and beta3GalT5 are cancer specific and significant markers for breast cancer stem cells. Proceedings of the National Academy of Sciences of the United States of America. 2016;113(4):960-5. Epub 2015/12/19.
Jacob F, Alam S, Konantz M, Liang CY, Kohler RS, Everest-Dass AV, et al. Transition of Mesenchymal and Epithelial Cancer Cells Depends on alpha1-4 Galactosyltransferase-Mediated Glycosphingolipids. Cancer research. 2018;78(11):2952-65. Epub 2018/03/25.
In other words, the intro is jumpy, and disconnected, there is no straight line narrative and that can be improved.
Point 6: Results: the authors have used a broad spectrum of concentrations of the drug, on both cell lines; it should therefore be possible to calculate a proper EC50 estimation.
Response 6: We added a precise IC50 values for both cell lines at the end of the section 2.1. and a sentence for IC50 calculation method in section 4.8. (now lines 161 – 162 and line 501).
Point 7: Apoptosis: I think the compounds are not very effective at inducing apoptosis, and they don't have to. So maybe this should not be overly discussed.
Response 7: Thank you for your comment. We rewrote a sentence discussing the induction of the apoptosis (line 174).
Point 8: Mammosphere formation: It's not clear what we gain from the mammosphere assay. It appears to me that the compound is simply blocking proliferation, either specifically or in a non-specific fashion, due to cytotoxicity. The mammosphere assay doesn't add anything to this since they are still growing... i.g., they are proliferating and the drug is clearly antiproliferative (but not so much pro-apoptotic). However, I think it would be much more interesting if the authors would take "mature", fully formed mammospheres, and see if the drug has any long(er) term effects on these aggregates. Simply, since there should be a (marginal, but measurable) enrichment of "stemness" in these spheroids and this may be targeted - as the authors claim - by the drug. They may also have a look at the cells in the centre of these spheroids, as these display the highest degree of "stemness", for whatever reasons.
Response 8: Thank you for your suggestion. Indeed, we cannot be sure whether the drug treatment caused a reduction in mammosphere formation by reducing the number of stem cells. However, we strongly assume that this was the case, as the shape and size of the mammospheres changed significantly after treatment, as we explained in section 2.3. In addition, the percentage of CD44+/CD24- cells observed in the parallel experiment after treatment was significantly reduced, which also confirms our assumption.
It would be great to do the experiment you suggested, but now it is quite inconvenient to do it. However, we were guided by the similar studies conducted to elucidate the effect of new potential drugs, and the mammosphere formation test was only one of the methods that contributed to our final conclusions.
Point 9: Cancer Stem Cells: the authors claim in Fig. 4 to look at stem cell populations in these 2 cell lines. I am not convinced these really are stem cells, and would expect additional proof. Are these cells responding to any estrogens? Growth factors (they shouldn't). Are they proliferating (they should be slowly). This requires more detailed analysis and also more explanation. The isolation and characterization of these cells from the 2 cell lines are barely described at all, certainly, no details whatsoever are provided in the results section. What's the ratio of these "stem cells" to luminally differentiated cells? The current paragraph and Fig. 4 cannot be accepted in this format. Any characterization is also missing in the materials & methods part. This is a major problem of the manuscript and MUST be fixed before it can be considered for publication.
Response 9: We analysed the percentage of CD44+/CD24- cells in the total cell population by flow cytometry as described in Materials and Methods, section 4.3. We did not specifically isolate stem cells by response to estrogens or growth factors or by proliferation assay. Instead, we stained all cells with anti-CD44 and anti-CD24 antibodies and gated CD44+/CD24- cells to distinguish a subpopulation of stem cells from non-stem cells, as described in many studies that we have also cited in the Introduction section, references 5 - 8. Figure 4 shows the percentage of these gated cells, CD44+/CD24-, in the total cell population.
Point 10: Fig. 5: the expression of gangliosides also makes more sense if the ratio of CSCs over "other" cells is shown. At the present moment, I also don't see any interesting and significant effects of the compound. And the small changes that we do see, I don't know what do we learn from those data? So, what's the point?
Response 10: As answered to your previous comment, we described CSC as CD44+/CD24-, and non-CSC as CD44+/CD24+, CD44-/CD24- and CD44-/CD24+, as written in Notes below the Figures 5 and 6.
The significant effects of the compound on glycosphingolipid expression are discussed in sections 2.5 and 2.6 of the Results and throughout the Discussion section, particularly from line 356. Although the treatment with the compound did not dramatically alter glycophingolipid expression on CSC and non-CSC, it is important to also highlight small changes in the membrane architecture of cancer cells that are related to the observed metabolic changes.
Point 11: Maybe the most solid data are shown in Fig. 9 and 10, plus table 1. I have the hunch the authors are more experts in this field than cell biology and cell-based models. But that's okay. Maybe the manuscript should be focused on these metabolic/metabolomic studies, and conclusions should be drawn from these data, not from (maybe) stem cells that aren't properly characterized. I also don't really see the link a) between the compound and stem cell biology, and b) between the compound and the gangliosides. That's really not convincing.
Response 11: We hope that we have succeeded in dispelling all your doubts and explaining the link between the compound and stem cell biology and compound and gangliosides by responding to your previous comments.
Reviewer 2 Report
The authors synthesized a compound and validated its effect on % of cancer stem cells and the expression of glycosphingolipids on CSCs and non-CSCs on breast cancer cell lines. Further they also investigated the metabolic profile of newly synthesized compound in breast cancer cell lines.
Concerns are:
1. What is the rationale of using 2 different breast cancer cell lines;
(ideally to make a constructive study - at least 2 TNBC and 2 non-TNBC cell lines should be utilized)
2. Most of the data are not statistically significant
3. It would have been better to compare the effect of new compound 1 with already existing FDA drug to demonstrate its superior effectiveness.
Author Response
Point 1: What is the rationale of using 2 different breast cancer cell lines;
(ideally to make a constructive study - at least 2 TNBC and 2 non-TNBC cell lines should be utilized)
Response 1: A non-TNBC cell line MCF -7 expresses estrogen, progesterone and androgen receptors, while the TNBC cell line MDA-MD -231 expresses only androgen receptors. This difference in the expression of the receptors is related to the different content of glyocosphingolipids in the membrane and the induction of different metabolic pathways. In this manuscript, we have investigated the effect of the newly synthesised compound on these two cell lines in addition to identifying a potential anticancer target in breast cancer cell lines and in particular TNBC, the most severe form of breast cancer.
Yes, we agree that it would be ideal to use 2 TNBC and 2 non-TNBC cell lines, but we did not have such a resource.
Point 2: Most of the data are not statistically significant
Response 2: The cytotoxic effect of the compound is very significant, especially for TNBC cell line MDA-MB-231, where IC50 after 48h is 2.082 µM, as shown on Figure 1. Furthermore, apoptosis rate is also enhanced after treatment with the compound in both cell line, whilst mammosphere formation is reduced after treatment suggesting that the compound effects particularly stem cell population. Although treatment with the compound did not dramatically alter glycophingolipid expression on CSC and non-CSC, it is important to also highlight small changes in the membrane architecture of cancer cells that are related to the observed metabolic changes.
Point 3: It would have been better to compare the effect of new compound 1 with already existing FDA drug to demonstrate its superior effectiveness.
Response 3: We have analyzed the effect of similar compound from the thieno-pyridine family in combination with paclitaxel on MDA-MB-231 cells and found that paclitaxel in combination with higher compound concentrations showed antagonistic effect.
Mastelić A, ÄŒikeš ÄŒulić V, Režić Mužinić N, Vuica-Ross M, Barker D, Leung EY, Reynisson J, Markotić A. Glycophenotype of breast and prostate cancer stem cells treated with thieno[2,3-b]pyridine anticancer compound. Drug Des Devel Ther. 2017 Mar 14;11:759-769. doi: 10.2147/DDDT.S121122. PMID: 28352152; PMCID: PMC5359006
Round 2
Reviewer 1 Report
The authors haven't added any experiments - but they have significantly changed the text, and they tried indeed to cover all of the aspects raised against the manuscript. But maybe this is sufficient.
Author Response
Point 1: The authors haven't added any experiments - but they have significantly changed the text, and they tried indeed to cover all of the aspects raised against the manuscript. But maybe this is sufficient.
Response 1: In principle we agree with the reviewer that more experimental data would be better. However, we believe that our explanation suffices.
Reviewer 2 Report
Figure resolution is poor and must be improved before considering for publication
Author Response
Point 1: Figure resolution is poor and must be improved before considering for publication
Response 1: We included figures with higher resolution in re-re-submitted manuscript and uploaded them in a separate file in the system